# Characterising Sex-Specific Metabolite Differences in New Zealand Geoduck (*Panopea zelandica*) Using LC-MS/MS Metabolomics

**DOI:** 10.3390/ani15060860

**Published:** 2025-03-17

**Authors:** Leonie Venter, Andrea C. Alfaro, Jeremie Zander Lindeque, Peet J. Jansen van Rensburg, Natalí J. Delorme, Norman L. C. Ragg, Leonardo N. Zamora

**Affiliations:** 1Aquaculture Biotechnology Research Group, Faculty of Health and Environmental Sciences, Auckland University of Technology, Private Bag 92006, Auckland 1142, New Zealand; andrea.alfaro@aut.ac.nz; 2Biomedical and Molecular Metabolism Research, Faculty of Natural and Agricultural Science, North-West University, Private Bag 1290, Potchefstroom 2520, South Africa; zander.lindeque@nwu.ac.za (J.Z.L.); peet.jansenvanrensburg@nwu.ac.za (P.J.J.v.R.); 3Cawthron Institute, Private Bag 2, Nelson 7042, New Zealand; natali.delorme@cawthron.org.nz (N.J.D.); norman.ragg@cawthron.org.nz (N.L.C.R.); leo.zamora@cawthron.org.nz (L.N.Z.)

**Keywords:** aquaculture, broodstock management, clam, female, geoduck, male, metabolomics, *Panopea zelandica*, sex identification, shellfish hatchery

## Abstract

The development of non-lethal sex identification techniques is needed to support broodstock management within geoduck clams. *Panopea zelandica* is placed forward as an emerging aquaculture species in New Zealand and, as with other species, cannot be accurately sexed prior to spawning, complicating production and subsequent population growth. In this study, liquid chromatography-tandem mass spectrometry (LC-MS/MS) metabolomics was used to analyse gill and muscle tissue samples from male and female *P. zelandica*. From the results, 17 metabolites were identified that significantly differed between the sexes. Herein, lipid biosynthesis is suggested in female clams to support reproductive functions, while carbohydrate pathways sustain sperm production in males. This first-time investigation provides valuable insights into potential sex bio-markers, presenting metabolomics as a non-lethal sex identification method that can be utilised to improve breeding strategies in geoduck aquaculture.

## 1. Introduction

The large, infaunal, geoduck clam, *Panopea zelandica* (Quoy and Gaimard, 1835), is found in subtidal areas around New Zealand [1], protected by the Quota Management System which establishes a total allowable commercial catch and ensures sustainable harvesting [2]. Mostly, these clams are harvested by divers for export as live, chilled and frozen animals to traditional South East Asia and Chinese markets [3]. The increasing demand for geoduck in the global market is contributing to the establishment of geoduck aquaculture sectors to avoid overexploitation of natural populations [4], something which has yet to be established as commercial development in New Zealand [5]. Worldwide, geoduck broodstock are predominantly harvested from fisheries for hatchery spawning purposes, making it difficult to accurately age the animals and obtain insight into their genetic diversity [6]. Furthermore, geoduck are not sexually dimorphic [7], with the external morphology of the males and females being indistinguishable from each other [8].

Sex can only be accurately determined once spawning takes place. However, during spawning events not all broodstock may be mature or spawn, causing highly skewed sex ratios in some populations [9]. Currently, the sex of geoduck can be determined via biopsies or microscopic observations of gonadal tissue after post-mortem dissections [8]. The goal is to find non-lethal sex identification methods in broodstock animals to manage captive populations and wild stock. Some progress has been made with the rapid identification of sex and/or maturation in geoduck, with vitellogenin, a precursor protein of egg yolk, found as a biomarker in female *Panopea japonica* using an enzyme-linked immunosorbent assay [8]. Another study on *Panopea generosa* highlights the complexity of sex identification, with pathways of steroid metabolism poorly understood [9], necessitating new research to investigate biochemical mechanisms and biological functions of steroidal metabolism in molluscs.

Omics approaches are powerful biomarker discovery tools enabling the study of complex interactions between genotypes and phenotypes, quantifying biological molecules (genes, proteins, and metabolites) to understand their structure, function, and dynamics within a biological system [10]. The use of transcriptome analyses of gonad tissues has proven successful in differentiating between male and female bivalves as seen amongst others in scallops [11], oysters [12], and clams [13]. Then again, metabolomics allows studying of the metabolome, providing insights into the functional state of cells and serving as a direct signature of biochemical activity [14]. The use of nuclear magnetic resonance metabolomics analyses to find sex-related differences has been successful in other invertebrate species. For example, in abalone, depleted sources of adenosine mono- and triphosphate along with betaine were found in males exposed to organotin compounds [15]. Male clams showed higher alanine and glycine and lower acetoacetate, choline and phosphocholine compared to females [16]. Additionally, metabolomics has revealed species-specific sex differences for both *Mytilus edulis* and *Mytilus galloprovincialis*, while also providing the best measure of functional reproductive status when using mantle tissue [17].

Considering the status of *Panopea zelandica* as high value species with farming potential [5], the use of new technologies to better understand the internal biochemical and/or physiological mechanisms involved in the reproductive processes [7], along with efforts to develop farming practices to increase seed numbers [8], are needed to boost this aquaculture sector. To this end, the aim of the current study was to apply liquid chromatography tandem mass spectrometry (LC-MS/MS) metabolomics as an investigation tool, to distinguish between metabolites relating to male and female *Panopea zelandica* gill and muscle samples collected from broodstock individuals housed in an experimental hatchery.

## 2. Materials and Methods

### 2.1. Animal Husbandry, Sampling and Sexing

Wild-sourced adult geoduck from the Golden Bay region (New Zealand) were collected and transferred for 120 km to the Cawthron Aquaculture Park. Animals were housed in 100 L tanks with flow-through seawater for 17 months (as part of an ongoing hatchery research program evaluating long term broodstock holding and multiple conditioning and spawning events). During this time, they were fed with standard hatchery grown microalgae species (e.g., *Chaetoceros muelleri* and *Tisochrysis lutea*—formerly *Isochrysis galbana*) aiming to maintain conditions and support gametogenesis until sampling took place in September (Spring in New Zealand).

Prior to sampling, a total of 23 geoduck were removed from the holding tanks, weighed to the nearest 0.01 g, and the shell lengths were measured to the nearest 0.10 mm along the longest axis using callipers. Geoduck were opened by inserting a scalpel above the mantle and cutting through the anterior and posterior adductor muscle attachments. A subsection of gill and anterior adductor muscle tissue was collected, placed in cryovials, snap-frozen using liquid nitrogen, and stored at −80 °C until metabolomics analyses were performed. Gills and muscle tissues were selected for analyses based on their distinct metabolic functions, with rapid metabolite turnover in the gills [18], while muscle tissue has slower protein synthesis and turnover allowing more stable metabolite detection [19]. Importantly, both organs lack reproductive tissue, which would naturally prejudice male/female differences.

To infer the sex and reproductive condition of the animals, a histological tissue section was collected. Using a dissection blade, a cross section of tissue including gonad, gill, digestive tract, digestive gland, heart, kidney, palp, nerve, musculature, mantle, and siphon was obtained for histological assessments. The histology samples were processed by a commercial laboratory for embedding, sectioning, and staining with haematoxylin and eosin. Histology slides were examined using a compound light microscope (Olympus BX40, Tokyo, Japan). Sex was determined by the presence of oocytes or spermatocytes under microscopic examination. All tissues were examined for abnormalities and the presence of any parasites and pathology.

From histological assessments, a total of 11 geoducks were selected after being graded as “good”, where none or a few general features (i.e., haemocyte aggregation, infiltration or ceroid, and pathogens) were found, as well as no tissue-specific issues present in the digestive tract (e.g., epithelium abnormalities), digestive gland (e.g., tubule sloughing), and mantle (e.g., density of connective tissue). The reproductive stage of the selected animals was classified as early active, late active, ripe, spawned, and spent, based on criteria by Gribben et al. (2004) [20]. Gill and muscle tissues from these 11 clams were then utilised for metabolomics analyses, in alignment with proposed replicate sampling suggested by the Metabolomics Standards Initiative [21].

### 2.2. Metabolomics Sample Preparation and Analysis

Stored tissue samples were freeze-dried overnight and then ground into fine powder using a mortar and pestle. Approximately 10 mg of ground tissue together with 20 μL of internal standard (10 mM L-alanine-2,3,3,3-d_4_) was extracted using a two-step methanol–water pre-blend extraction solvent mixture [1], resulting in extracts for liquid chromatography-tandem mass spectrometry (LC-MS/MS) analyses. Quality control (QC) samples were included in each batch to measure repeatability and to identify any potential batch effects in the data. The QC samples were prepared by pooling a mixture of either muscle or gill tissue and analysing them as a biological sample. An Agilent 1260 LC coupled to an Agilent 6470 triple quadrupole (QQQ) mass spectrometer (Agilent Technologies, Santa Clara, CA, USA) was used for metabolomics analyses. Agilent MassHunter Workstation Data Acquisition (V 10.0) was used for compound calibration and data acquisition. The LC-MS/MS data were pre-processed with an Agilent MassHunter Workstation QQQ Quantitative Analysis Software (V 10.0) [22]. Two unique transitions were monitored per individual metabolite to provide spectral matching in addition to retention time, resulting in metabolite identities with the highest level of confidence [21,23]. Data were normalised using the mass spectrometry total useful signal normalisation method and generalised log transformed [22].

Metabolite differences between male and female clams were detected via the online webserver MetaboAnalyst (https://www.metaboanalyst.ca, accessed on 2 December 2024), focusing on the findings obtained from the anterior adductor muscle and gill tissues separately [24]. Batch effects were verified using the QC samples by calculating the coefficient of variance percentages. Univariate analyses were used to determine statistically significant metabolites between male and female clams [*p*-value < 0.05 (false discovery rate ≤ 0.1)], while the effect size was calculated to ensure practical significance (d-value > 0.8, calculated by determining the absolute difference between the means of the two groups divided by the maximum standard deviation of the two groups) [25]. The metabolites of significance were further grouped by tissue-specific findings in a Venn diagram. Multivariate analyses were utilised to provide an overview of the metabolic changes and covariance. The average metabolite abundance of significant metabolites between male and female clams was visualised in a heatmap with metabolite clustering. Principle component analysis (PCA) was used to visualise the major trends between male and female clams by plotting ellipses with a 95% confidence level to effectively indicate grouping [26]. Additionally, a schematic representation of the overall metabolite response with the average (±SE) metabolite abundance detected in male and female clams was manually generated using the significantly detected metabolites.

## 3. Results

From the dissected clams, a total of seven males were present with an average (± SD) wet weight of 405.43 ± 74.29 g and shell length of 109.57 ± 8.81 mm, and reproductive stages were from late active to spent. The remaining four clams were females with an average (± SD) wet weight of 407.50 ± 35.49 g and shell length of 110.25 ± 6.60 mm, and reproductive stages were from spawned to spent.

LC-MS/MS analyses of gill and muscle tissues resulted in a total of 159 detected metabolites, of which the nine metabolites detected in the gill and ten in the muscle showed significant concentration differences between male and female samples, as determined by cut-off values (*p* < 0.05; d > 0.8). Of the significantly different metabolites, two metabolites (D-sedoheptulose-7-P and malic acid) were detected in both gill and muscle tissues, while an additional seven metabolites were detected in the gills only and eight metabolites detected in the muscle tissue only (Figure 1a). From the seventeen significant metabolites, four [cis-aconitic acid (gill), citric acid (gill), malic acid (gill), and malic acid (muscle)] showed higher concentrations in female geoduck compared to male geoduck, while the remaining metabolites were lower within the female clams (as depicted in the heatmap of Figure 1b). The PCA score plots show a clear separation between male (●) and female (■) metabolites of significance (Figure 1c). An overview of the male and female geoduck metabolite response (Figure 2) encompasses a range of metabolite classes including benzoic acid derivatives, purine and pyrimidine metabolites, tricarboxylic acid cycle metabolites, amino acids, carbohydrates, fatty acid conjugates, hydroxy acids, and organosulfonic acids (Table 1).

## 4. Discussion

Male and female animals differ in their physiology [27], yet geoduck show no sexual dimorphism and remain similarly sized [6], with sex only accurately identified due to the presence of spermatozoa or oocytes via histological assessment or through spawning induction [8]. This study investigated, for the first time, metabolite changes between male and female *P. zelandica*, analysing anterior adductor muscle and gill tissues via LC-MS/MS metabolomics. The results showed differences in metabolite levels between male and female geoduck, with females showing mostly a reduced response in metabolite abundance. In *P. zelandica*, a sperm to egg ratio of ≤100:1 has been reported to secure successful fertilisation [28], which might necessitate increased metabolite concentrations in muscle and gill tissues of male clams to support spermatozoa energy availability [29]. Considering that the cost of reproduction is believed to be higher in females than in males (considering the large egg size compared to sperm) [30,31], less metabolic energy via gill and muscle sources is suggested in female clams. When a large energy allocation goes to reproduction (i.e., gonad development), a reduction in available aerobic scope is seen [32], supporting the reduced metabolite concentrations detected in adjacent tissues of the female geoduck under investigation.

The metabolites, malic acid (in both gill and muscle tissue), citric acid, and cis-aconitic acid, showed higher concentrations in female geoduck (compared to the males), which are linked directly to the tricarboxylic acid (TCA) cycle, the central process of energy metabolism [33]. The TCA cycle utilises acetyl-coenzyme A (-CoA), derived from glucose via the glycolytic pathway, or fatty acids via beta (β)-oxidation, to produce citric acid [34]. During the next steps of the cycle, isocitric acid, alpha (α)-ketoglutaric acid, succinyl-CoA, succinic acid, malic acid, and oxaloacetic acid are formed, providing reducing equivalents that feed the electron transport chain [35,36]. Interestingly, in the cytosol, citric acid can be broken down to oxaloacetic acid (and malic acid which re-enters the TCA cycle) and acetyl-CoA is processed to malonyl-CoA, which in turn supports fatty acid synthesis [34]. It is hypothesised that female geoduck enhance acetyl-CoA production to support lipid biosynthesis, attributed to the increased citric acid and malic acid detected in this study. Increased lipid metabolism has been documented in studies profiling female mice [37], algae [38], and humans [39] as a result of increased hormones linked to reproductive functions. In a study on the clam, *Mactra chinensis*, the lipid content calculated from adductor muscle samples between males and females did not differ between seasons [40], nor were there any clear differences in haemolymph lipid concentrations between male and female geoduck, *P. globosa* [7]. In the mussel, *Mytilus galloprovincialis*, lipid metabolism was more impacted in female than male mussels following exposure to anti-depressant drugs [41]. Overall profiling of fatty acid content within male and female *P. zelandica* will benefit future studies aimed at sex-characterisation.

It is well known that the production of gametes is an energetically costly process, necessitating stored energy in the form of dietary proteins, lipids and carbohydrates [7]. Findings from the current study suggest that male geoduck showed an increase in carbohydrate-based metabolites (D-sedoheptulose-7-phosphate and glucoheptonic acid), likely to support the formation of spermatozoa membrane lipids, which do not accumulate lipid reserves but rather utilise carbohydrates to support motility, as reported in blue mussels [42]. Also, the input of metabolites toward the glycolysis pathway could potentially be lower in the females under investigation as females utilise glycogen during conditioning [43]. Moreover, lactic acid was increased in male geoduck as a result of increased production or decreased clearance thereof. Typically, lactic acid accumulates when energy demand exceeds the supply of oxidative metabolism [44], for example, due to an increased demand, as seen in muscles of the burrowing bivalve *Paphies subtriangulatum* [45]. Herein, the increased lactic acid in the muscle tissue of male geoduck can be seen as an added pathway to maintain metabolic activity [46] and to support glucose metabolism [47]. It was reported that the adductor muscle of *P. generosa* does not attribute towards storage functions [7], yet this tissue can still support the active use of metabolites for physiological functions as hypothesised in this study.

Other metabolites, such as aminoadipic acid, methionine, and taurine, were also increased in male geoduck, highlighting the added energy requirements to support gonadal development as seen in male oysters [48]. For instance, aminoadipic acid (derived from lysine metabolism) plays an important role in protein synthesis [49], while proteins coding for methionine have been highlighted as sex-specific in sea cucumbers [50]. In contrast, taurine is not involved in protein synthesis but generally occurs in high intracellular concentrations in marine organisms [51] and has been highlighted as a biomarker to distinguish between male and female mussels [52] and zebrafish [53]. In the geoduck under investigation, males showed higher concentrations of taurine than females, potentially to regulate hormone levels, as stated in sea cucumbers [54], and to support osmoregulation, as seen in salmon [55] and oysters [56].

The process of gametogenesis is known to require substantial purine and pyrimidine investment to produce sperm and oocytes [57], as confirmed in the current study where purine (dGMP, cAMP, ATP, and ITP) and pyrimidine (deoxycytidine) metabolites significantly differed between male and female geoduck. Broadly, purine metabolites can be converted into ATP as a storage source and to cAMP to serve as a messenger molecule to regulate metabolism, as reported in sex-related genes of clams, where purine metabolism dominated in males [58]. Enrichment of purine metabolism pathways was also seen in male scallops [59]. In oysters, purine metabolism was said to be involved in the maintenance of oocyte maturation [60], and in female zebrafish, the depletion of the precursor phosphoribosyl pyrophosphate resulted in reduced intermediates within purine metabolism [53]. Conversely, in female mussels, pyrimidine metabolism was down-regulated in comparison with males following exposure to a stressor [41]. The male geoduck under investigation showed an increase in purine and pyrimidine metabolites (compared to the females), supporting the research mentioned above. It is hypothesised that male geoduck depend on the role of purine metabolism in energy supply [61] to fulfil increased energy needs.

As a rule, differentiation between sexes on a physiological level (i.e., energetics and metabolism) is driven by various factors, such as hormone levels [62], sex steroids [63], biochemical signatures from egg and sperm [64], and sex chromosomes [65]. Within the current investigation, none of the above factors were directly measured, yet sex-specific variations in metabolites were detected between male and female *P. zelandica*. This study indicates that metabolites associated with lipid biosynthesis were increased in female clams, while all other affected metabolites were decreased in the female cohort. The results additionally support an increase in carbohydrate-associated metabolic pathways in male geoduck, arguably meant to sustain sperm production. Taurine has been reported as a biomarker to distinguish between sexes in bivalves and is confirmed within this study as a metabolite to distinguish between male and female geoduck. Moreover, male geoduck had increased purine and pyrimidine biosynthesis, supporting energy needs.

## 5. Conclusions and Future Recommendations

This study provides advancements in geoduck sex identification on a metabolite level using muscle and gill tissues. Considering that geoduck, lack clear sexual dimorphism, the application of biomarker approaches, such as metabolomics, to infer sex differences becomes a useful tool within a wider aquaculture context [66]. As a next step, a non-destructive biological sample, such as haemolymph, can be analysed to determine if the same metabolites can be reliably detected [67]. Also, future studies will benefit from targeting lipid metabolism, as proven by research on *Apostichopus japonicus* [54] and *Odontobutis potamophila* [68], as well as phospholipid metabolism seen in *Perna viridis* [69]. The use of gonads and digestive gland as tissues to infer meaning to sex differences has also proven productive [69] and is worth considering for future metabolomics studies on geoduck sex characterisation. It is also important to include information relating to gonad development to allow the tracking of gametogenesis, which can assist biomarker development for different developmental stages [70]. Ultimately, sex-specific omics responses from geoduck tissues are lacking in the literature; hence, this study creates a new dataset that can contribute to the characterisation of male and female geoduck. This information provides valuable references for future breeding of *P. zelandica* in New Zealand.

## Figures and Tables

**Figure 1 animals-15-00860-f001:**
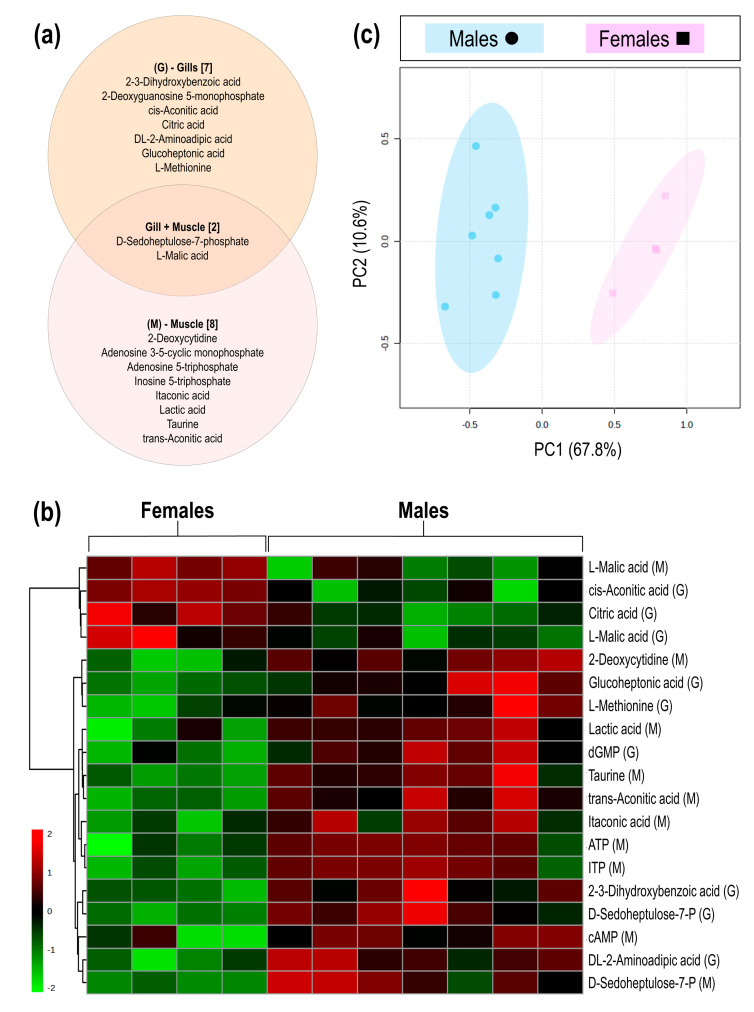
The metabolite differences between male and female geoduck, *Panopea zelandica*: (**a**) Venn diagram of metabolites detected in gill and muscle tissues, (**b**) Heatmap visualisation of significant metabolites detected in gill (G) and muscle (M) tissue of each individual (column), and (**c**) PCA score plot of the differential metabolite grouped by sex.

**Figure 2 animals-15-00860-f002:**
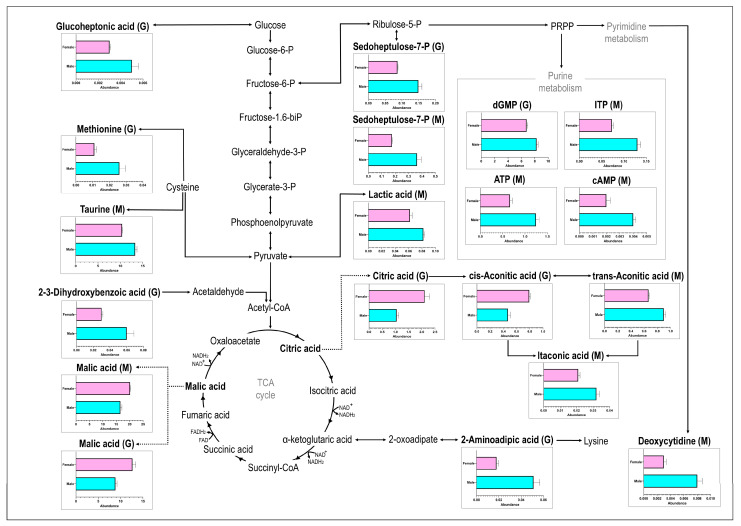
The metabolite response of female (top bar in pink) and male (bottom bar in blue) geoduck, *Panopea zelandica*, detected in gill (G) and muscle (M) tissue, reported as average metabolite abundance distributed amongst different metabolite pathways.

**Table 1 animals-15-00860-t001:** Metabolites listed as significantly lower (↓) or higher (↑) in female geoduck, *Panopea zelandica*, compared to their male counterparts, as detected in gill or muscle tissue. Also reported are the cut-off values (d > 0.8, *p* < 0.05), a Kyoto Encyclopaedia of Genes and Genomes (KEGG) identification number, and a metabolite class grouping.

Compound	Female Response	Tissue	d-Value	*p*-Value	KEGG ID	Metabolite Class
2,3-Dihydroxybenzoic acid	↓	Gills	1.22	6.45 × 10^−3^	C00196	Benzoic acids and derivatives
2-Deoxycytidine	↓	Muscle	2.18	9.87 × 10^−4^	C00881	Pyrimidine 2′-deoxyribonucleosides
2-Deoxyguanosine 5-monophosphate (dGMP)	↓	Gills	2.11	4.52 × 10^−3^	C00362	Purine deoxyribonucleotides
Adenosine 3-5-cyclic monophosphate (cAMP)	↓	Muscle	1.52	1.24 × 10^−2^	C00575	Cyclic purine nucleotides
Adenosine 5-triphosphate (ATP)	↓	Muscle	2.56	2.95 × 10^−3^	C00002	Purine ribonucleotides
cis-Aconitic acid	↑	Gills	2.69	4.26 × 10^−3^	C00417	Tricarboxylic acids and derivatives
Citric acid	↑	Gills	1.90	1.97 × 10^−3^	C00158	Tricarboxylic acids and derivatives
DL-2-Aminoadipic acid	↓	Gills	1.85	1.34 × 10^−3^	C00956	Amino acids, peptides, and analogues
D-Sedoheptulose-7-phosphate	↓	Gills	1.82	2.47 × 10^−3^	C05382	Carbohydrates and carbohydrate conjugates
D-Sedoheptulose-7-phosphate	↓	Muscle	1.85	9.31 × 10^−4^	C05382	Carbohydrates and carbohydrate conjugates
Glucoheptonic acid	↓	Gills	1.33	6.46 × 10^−3^	NA	Carbohydrates and carbohydrate conjugates
Inosine 5-triphosphate (ITP)	↓	Muscle	2.40	1.35 × 10^−3^	C00081	Purine ribonucleotides
Itaconic acid	↓	Muscle	1.72	8.87 × 10^−3^	C00490	Fatty acids and conjugates
Lactic acid	↓	Muscle	1.90	2.80 × 10^−3^	C00186	Alpha hydroxy acids and derivatives
L-Malic acid	↑	Gills	1.60	5.33 × 10^−3^	C00149	Beta hydroxy acids and derivatives
L-Malic acid	↑	Muscle	2.01	6.29 × 10^−3^	C00149	Beta hydroxy acids and derivatives
L-Methionine	↓	Gills	1.29	7.32 × 10^−3^	C00073	Amino acids, peptides, and analogues
Taurine	↓	Muscle	2.30	8.65 × 10^−4^	C00245	Organosulfonic acids and derivatives
trans-Aconitic acid	↓	Muscle	2.41	5.92 × 10^−4^	C02341	Tricarboxylic acids and derivatives

## Data Availability

Raw metabolomics data are provided as Appendix A with this manuscript. Further inquiries can be directed to the corresponding author.

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
