# Peer review of "Characterising Sex-Specific Metabolite Differences in New Zealand Geoduck (Panopea zelandica) Using LC-MS/MS Metabolomics"

_animals, 2025, doi:10.3390/ani15060860_

Round 1

Reviewer 1 Report

Comments and Suggestions for Authors

the ms provided a new dataset contributing to the characterisation of male and female geoduck. the results are helpful to distinguish sex in the other shellfish. the ms is well written and some suggestions for minor revisions are liseted as follows.
to add the recent literature on gonad  in the shellfish by omics analysis in the introduction.
to add the results on the stage of gonad developmentï¼›growth traits involved in shell length and shell height were provived in the ms. also, to add the values of gonad index. 
to add histological assessments. 

Author Response

Reviewer 1:

The ms provided a new dataset contributing to the characterisation of male and female geoduck. the results are helpful to distinguish sex in the other shellfish. the ms is well written and some suggestions for minor revisions are listed as follows:

Comment: Add the recent literature on gonad in the shellfish by omics analysis in the introduction.

Response: Thank you for the suggestion. Additional research concerning gonad assessments via other omics approaches has been added to the introduction section.

Comment: Add the results on the stage of gonad development; growth traits involved in shell length and shell height were provided in the ms.

Response: Results of the gonad maturation staging as well as the criteria used have been added to the methods and results sections respectively, as requested by the reviewer.

Comment: Add the values of gonad index.

Response: Unfortunately, gonad index values were not determined within the original assessments performed in this study. It is, however, added as an important aspect to be considered for future work aimed at sex-identification.

Comment: Add histological assessments.

Response: Criteria used to identify and select “good” geoducks used during histological assessments have been added to the manuscript. Comprehensive histological assessments of these individuals fall outside the scope of the current study and are submitted for publication elsewhere.

Reviewer 2 Report

Comments and Suggestions for Authors

After a detailed analysis of the manuscript entitled "Characterizing sex-specific metabolite differences in New Zealand geoduck (Panopea zelandica) using LC-MS/MS metabolomics", by Venter et al., it was possible to identify important contributions, such as the identification of sex biomarkers through metabolomics, which can optimize geoduck aquaculture production and provide a more effective response to the growing global demand. However, some points require reformulation. Below, I present general comments and textual issues that should be resolved before publication.

- The second sentence of the first paragraph of the discussion (lines 219-222) is out of context. It could be reformulated as a conclusion or, alternatively, inserted as part of an introductory paragraph to the discussion.

- I recommend inserting a reference in the first sentence of the first paragraph of the discussion.

- To improve clarity, cohesion and fluidity, I suggest a general restructuring of the first two paragraphs of the discussion. I had to reread these sections four times to properly understand what the authors were trying to convey, which is not ideal for the reader.

- I suggest including an explanatory table on the function of the metabolites observed in P. zelandica, as well as an indication of whether these metabolites have already been used as biomarkers of sexual differentiation in other organisms.

- I suggest including an image of the male and female organisms to reinforce the claim that there is no apparent sexual dimorphism. In addition, it would be useful to present histological images of sexing by microscopic examination, indicating the presence of oocytes or spermatocytes.

In view of these points, I suggest that the manuscript be revised, taking into account the suggestions above. I appreciate the opportunity to evaluate the work, and I hope that this review contributes to the improvement of the manuscript.

Author Response

Reviewer 2:

After a detailed analysis of the manuscript entitled "Characterizing sex-specific metabolite differences in New Zealand geoduck (Panopea zelandica) using LC-MS/MS metabolomics", by Venter et al., it was possible to identify important contributions, such as the identification of sex biomarkers through metabolomics, which can optimize geoduck aquaculture production and provide a more effective response to the growing global demand. However, some points require reformulation. Below, I present general comments and textual issues that should be resolved before publication:

Comment: The second sentence of the first paragraph of the discussion (lines 219-222) is out of context. It could be reformulated as a conclusion or, alternatively, inserted as part of an introductory paragraph to the discussion.

Response: This comment can be regarded as an author-specific preference. It is common for studies to mention the aim of the study in the discussion section again, and repeat the analyses performed. This style of discussion is found within various manuscripts published within the journal Animals. The sentence is retained in the discussion as per the author’s writing style.

Comment: I recommend inserting a reference in the first sentence of the first paragraph of the discussion.

Response: Thank you for the recommendation, references have been added to support this sentence.

Comment: To improve clarity, cohesion and fluidity, I suggest a general restructuring of the first two paragraphs of the discussion. I had to reread these sections four times to properly understand what the authors were trying to convey, which is not ideal for the reader.

Response: As authors we respectfully disagree with the reviewer in this regard. We have carefully reassessed the paragraphs and made minor changes to ensure clarity. The discussion type followed is commonly used to convey metabolomics findings, and herein the metabolite response of female geoduck is being discussed following the flow of energy. Moreover, the other reviewers did not comment on the flow of the discussion.

Comment: I suggest including an explanatory table on the function of the metabolites observed in P. zelandica, as well as an indication of whether these metabolites have already been used as biomarkers of sexual differentiation in other organisms.

Response: It is not possible to include an explanatory table on the function of the metabolites observed in P. zelandica, as metabolites have various functions within a complex metabolism network. The roles and concentrations of metabolites in a biological system are constantly changing and adapting depending on the organism’s physiological state, environmental factors, etc. It would be highly irresponsible to confine the metabolite functions to a table in this manuscript. The functions suggested in-text within the discussion are based on supporting literature and written in such a way to support the view of possible scenarios or hypotheses. Also, the focus here was more on bioenergetics and the contribution different pathways make in the different sexes (towards bioenergetics) (Mauvais-Jarvis, F. (2024). Sex differences in energy metabolism: natural selection, mechanisms and consequences. Nature Reviews Nephrology, 20(1), 56-69.).

Comment: I suggest including an image of the male and female organisms to reinforce the claim that there is no apparent sexual dimorphism. In addition, it would be useful to present histological images of sexing by microscopic examination, indicating the presence of oocytes or spermatocytes.

Response: Male and female geoduck look identical at face value ~ as for many other marine invertebrate species lacking sexual dimorphism, it would add no value to this manuscript to add images of this species. As mentioned within the comments to reviewer 1, the suggestion to add data or images from histological assessments goes beyond the scope of this manuscript and unfortunately cannot be included.

In view of these points, I suggest that the manuscript be revised, taking into account the suggestions above. I appreciate the opportunity to evaluate the work, and I hope that this review contributes to the improvement of the manuscript.

Reviewer 3 Report

Comments and Suggestions for Authors

The manuscript titled "Characterising sex-specific metabolite differences in New Zealand geoduck (Panopea zelandica) using LC-MS/MS metabolomics" provides a novel exploration of sex-related metabolic pathways. The manuscript successfully identifies 17 metabolites with significant sex-specific differences across gill and muscle tissues, including tissue-specific variations such as elevated taurine in male adductor muscle and lipid biosynthesis markers in female tissues. These findings align with established reproductive energetics in bivalves, where lipid reserves typically support oogenesis in females, while carbohydrate and nucleotide metabolism fuels spermatogenesis in males. Overall, the paper is well-written, informative, and presents a thorough synthesis of existing literature, but I have a few questions and suggestions: 

1.Only 11 out of 23 geoducks were classified as "healthy" based on histology, but the criteria used to exclude are unclear. The low inclusion rate (48%) may raises concerns about potential biases in the final dataset.
2.Why were gill and muscle tissues selected instead of the mantle for metabolomic analyses?
3.The reason for maintaining geoducks for 17 months prior to sampling?
4.Were cumulative thermal units tracked during the rearing period to quantify their relationship with gonadal development? 
5.The study does not specify the season or month of sampling.
6.Most bivalve species exhibit clear sexual dimorphism through visual. A discussion clarifying the niche utility of this approach within the broader context of bivalve aquaculture would strengthen the translational impact of the study.

Author Response

Reviewer 3:

The manuscript titled "Characterising sex-specific metabolite differences in New Zealand geoduck (Panopea zelandica) using LC-MS/MS metabolomics" provides a novel exploration of sex-related metabolic pathways. The manuscript successfully identifies 17 metabolites with significant sex-specific differences across gill and muscle tissues, including tissue-specific variations such as elevated taurine in male adductor muscle and lipid biosynthesis markers in female tissues. These findings align with established reproductive energetics in bivalves, where lipid reserves typically support oogenesis in females, while carbohydrate and nucleotide metabolism fuels spermatogenesis in males. Overall, the paper is well-written, informative, and presents a thorough synthesis of existing literature, but I have a few questions and suggestions:

Comment: Only 11 out of 23 geoducks were classified as "healthy" based on histology, but the criteria used to exclude are unclear. The low inclusion rate (48%) may raise concerns about potential biases in the final dataset.

Response: We agree with the reviewer’s perspective in this regard and have added additional caution to the methods section of this manuscript. Even though the number of replicates used in this study appears low, this falls into the proposed replicate sampling to be used as suggested by the Metabolomics Standards Initiative (Sumner, L. W., Amberg, A., Barrett, D., Beale, M. H., Beger, R., Daykin, C. A., ... & Viant, M. R. (2007). Proposed minimum reporting standards for chemical analysis: chemical analysis working group (CAWG) metabolomics standards initiative (MSI). Metabolomics, 3, 211-221).

Comment: Why were gill and muscle tissues selected instead of the mantle for metabolomic analyses?

Response: We do comprehend that these are not sex-specific organs and have made clear comments about them within the methods and suggestions for future studies. Ultimately, this was opportunistic sampling with the muscle and gill being the only available tissues for analyses. Yet, both tissues perform important metabolite functions as outlined in the manuscript.

Comment: The reason for maintaining geoducks for 17 months prior to sampling?

Response: These animals were acquired as broodstock for hatchery technology development, including long-term broodstock holding in captivity (e.g., 17 months) and multiple conditioning and spawning events. This has been identified as a key step towards domestication for this iconic shellfish species. We now expand on this in the first part of the methods section.

Comment: Were cumulative thermal units tracked during the rearing period to quantify their relationship with gonadal development?

Response: Unfortunately, no. Standard husbandry monitoring including temperature checks were conducted during the time the animals were in captivity, but no abnormal thermal events were documented.

Comment: The study does not specify the season or month of sampling.

Response: Both sampling month and seasons have now been added to the manuscript.

Comment: Most bivalve species exhibit clear sexual dimorphism through visual. A discussion clarifying the niche utility of this approach within the broader context of bivalve aquaculture would strengthen the translational impact of the study.

Response: Indeed, this perspective is now included in the manuscript.